# Identification of Tumor-Suppressive *miR-30a-3p* Controlled Genes: *ANLN* as a Therapeutic Target in Breast Cancer

**DOI:** 10.3390/ncrna10060060

**Published:** 2024-11-30

**Authors:** Reiko Mitsueda, Ayako Nagata, Hiroko Toda, Yuya Tomioka, Ryutaro Yasudome, Mayuko Kato, Yoshiaki Shinden, Akihiro Nakajo, Naohiko Seki

**Affiliations:** 1Department of Breast and Thyroid Surgery, Graduate School of Medical and Dental Sciences, Kagoshima University, Kagoshima 890-8520, Japan; k7854142@kadai.jp (R.M.); k8235150@kadai.jp (A.N.); k0848362@kadai.jp (H.T.); k7682205@kadai.jp (R.Y.); k4483271@kadai.jp (Y.S.); k4814560@kadai.jp (A.N.); 2Department of Pulmonary Medicine, Graduate School of Medical and Dental Sciences, Kagoshima University, Kagoshima 890-8520, Japan; k4829264@kadai.jp; 3Department of Functional Genomics, Chiba University Graduate School of Medicine, Chiba 260-8670, Japan; mayukokato@chiba-u.jp

**Keywords:** breast cancer, microRNA, *miR-30a-3p*, passenger strand, *ANLN*

## Abstract

Our recently created RNA-sequence-based microRNA (miRNA) expression signature in breast cancer clinical specimens revealed that some *miR-30* family members were significantly downregulated in cancer tissues. Based on TCGA database analyses, we observed that among the *miR-30* family members, *miR-30a-3p* (the passenger strand derived from pre-*miR-30a*) was significantly downregulated in breast cancer (BC) clinical specimens, and its low expression predicted worse prognoses. Ectopic expression assays showed that *miR-30a-3p* transfected cancer cells (MDA-MB-157 and MDA-MB-231) had their aggressive phenotypes significantly suppressed, e.g., their proliferation, migration, and invasion abilities. These data indicated that *miR-30a-3p* acted as a tumor-suppressive miRNA in BC cells. Our subsequent search for *miR-30a-3p* controlled molecular networks in BC cells yielded a total of 189 genes. Notably, among those 189 genes, cell-cycle-related genes (*ANLN*, *MKI67*, *CCNB1*, *NCAPG*, *ZWINT*, *E2F7*, *PDS5A*, *RIF1*, *BIRC5*, *MAD2L1*, *CACUL1*, *KIF23*, *UBE2S*, *EML4*, *SEPT10*, *CLTC*, and *PCNP*) were enriched according to a GeneCodis 4 database analysis. Moreover, the overexpression of four genes (*ANLN*, *CCNB1*, *BIRC5*, and *KIF23*) significantly predicted worse prognoses for patients with BC according to TCGA analyses. Finally, our assays demonstrated that the overexpression of *ANLN* had cancer-promoting functions in BC cells. The involvement of *miR-30a-3p* (the passenger strand) in BC molecular pathogenesis is a new concept in cancer research, and the outcomes of our study strongly indicate the importance of analyzing passenger strands of miRNAs in BC cells.

## 1. Introduction

Breast cancer (BC) is the most common malignant cancer and the leading cause of cancer-related deaths in women, with approximately 2.3 million new cases and 685,000 deaths reported worldwide in 2020 [1]. It is important to note that the incidence of breast cancer is rapidly increasing in Asian countries, including Japan [2]. The underlying reasons for this are changes in the lifestyles of Asian women, such as low birth rates, an increased prevalence of overweight and obesity, and decreased physical activity [3].

BC is histologically classified into four subtypes based on the expression levels of the estrogen receptor (ER), progesterone receptor (PR), and human epidermal growth factor receptor 2 (HER2), i.e., luminal A, luminal B, triple-negative (TNBC), and HER2-positive [4,5,6]. Essentially, treatment plans are created for BC patients based on these classifications [4,6]. As a good example, the advent of anti-HER2 targeted therapy has significantly improved the prognosis of HER2-positive breast cancer, which previously had a poor prognosis [7,8].

Metastasis is an event that has a significant impact on the prognosis of BC patients. Brain metastases are a common site of disease progression and represent a significant challenge in the treatment of patients with metastatic BC [9]. It has been reported that the probability of brain metastasis varies greatly depending on the patient’s subtype. Approximately one-third of patients with metastatic HER2-positive BC or TNBC develop brain metastases [10].

Several molecular assays based on multigene signatures, such as Oncotype DX, MammaPrint, Prosigna, and EndoPredict, have been developed to predict the risk of recurrence in early BC [7]. The search for genome-wide prognostic markers will accelerate the molecular diagnosis of BC patients.

In the post-Human Genome Project era, the existence of large amounts of non-coding RNAs in the human genome has become clear, and analyses and understanding that consider non-coding RNAs have become essential for cancer research. MicroRNAs (miRNAs) are small-length non-coding RNAs that bind 3′-untranslated regions of target messenger RNAs (mRNAs) in a sequence-dependent manner and function as controllers of gene expression at the post-transcriptional level [11].

The expression of most genes in human cells is controlled by miRNAs, and their expression control is involved in a variety of biological processes, e.g., cell cycle control, programmed cell death, differentiation, and invasiveness [12]. A single miRNA controls the expression of a huge number of mRNAs within a cell. Therefore, the aberrant expression of miRNAs leads to the disruption of tightly controlled intracellular RNA networks and is closely related to the development of human cancer.

We generated an RNA-sequence-based miRNA expression signature of BC to identify tumor-suppressive miRNAs and their controlled BC-promoting genes [13,14]. Notably, our signature revealed that some passenger strand miRNAs derived from pre-miRNAs were significantly downregulated in BC tissues [14,15]. According to the previous concept of miRNA biogenesis, two types of mature miRNAs are derived from pre-miRNAs. One strand (the guide strand) is selected for loading into the miRNA-Induced Silencing Complex (miRISC). The miRISC (including the guide strand) targets mRNAs for silencing in a sequence-dependent manner. On the contrary, the passenger strand miRNAs (the other strand of pre-miRNA) are thought to be degraded in the cytoplasm and have previously been considered nonfunctional [16]. However, accumulating evidence suggests that some passenger strands of miRNAs act as oncogenes and tumor-suppressive miRNAs through controlling their target genes [12,14].

Research into passenger strand-focused miRNAs can lead to the discovery of new molecular pathways and therapeutic targets for BC. Based on our miRNA signature of BC, we focused on the *miR-30* family, whose expression was suppressed in BC tissues. We have continued to explore the antitumor functions of *miR-30* family members *(miR-30c-1-3p* and *miR-30c-2-3p*) and their target molecules in BC [13,14]. In this study, *miR-30a-3p* (the passenger strand derived from pre-*miR-30a*) was focused on, and its functional significance and targets in BC cells were examined. Our present data demonstrate that *miR-30a-3p* acts as a tumor-suppressive miRNA in BC cells through controlling the following cell-cycle-related genes: *ANLN*, *MKI67*, *CCNB1*, *NCAPG*, *ZWINT*, *E2F7*, *PDS5A*, *RIF1*, *BIRC5*, *MAD2L1*, *CACUL1*, *KIF23*, *UBE2S*, *EML4*, *SEPT10*, *CLTC*, and *PCNP*. Moreover, four genes (*ANLN*, *CCNB1*, *BIRC5*, and *KIF23*) were good prognostic markers for BC patients and therapeutic target molecules.

## 2. Results

### 2.1. Genomic Structure of miR-30a-5p and miR-30a-3p, and Their Expression in BC Clinical Specimens

We previously created an miRNA expression signature of BC based on RNA sequencing [13]. Our signature revealed that both strands of pre-*miR-30a* (*miR-30a-5p*: the guide strand; *miR-30a-3p*: the passenger strand) were downregulated in BC tissues (Figure 1A).

The human genome database showed that pre-*miR-30a* was located on chromosome 6q13. Notably, two miRNAs (*miR-30a* and *miR-30c-2*) were located in close proximity in this region (Figure 1B). Among these miRNAs, the downregulation of *miR-30a-3p* and *miR-30c-2-3p* was detected via a TCGA-BRCA database analysis. The expression level of *miR-30a-3p* was significantly reduced in BC tissues, whereas *miR-30a-5p* exhibited no significant difference, as confirmed by the large amount of cohort data from the TCGA-BRCA datasets (Figure 1C). However, low expression levels of *miR-30a-5p* and *miR-30a-3p* were associated with a significantly lower 10-year overall survival rate than that associated with high expression levels of these miRNAs (Figure 1D). A Spearman’s rank analysis revealed a positive correlation between the expression levels of *miR-30a-5p* and *miR-30a-3p* (*r* = 0.833, *p* < 0.001; Figure 1E).

Furthermore, we investigated the expression levels of *miR-30a-5p* and *miR-30a-3p* according to BC subtypes, i.e., luminal, HER2-positive, and TNBC. The expression level of *miR-30a-5p* was significantly decreased in HER2-positive BC and TNBC compared with normal breast tissue (Appendix A). In contrast, a decreased expression of *miR-30a-3p* was confirmed in all subtypes (Appendix A).

### 2.2. Antitumor Roles of miR-30a-3p in BC Cells

To evaluate the antitumor roles of m*iR-30a-3p*, we applied ectopic expression assays to BC cell lines (MDA-MB-157 and MDA-MB-231).

The expression of *miR-30a-3p* inhibited the proliferation of BC cells (Figure 2A), and cancer cell invasion and migration abilities were markedly suppressed after *miR-30a-3p* expression in BC cells (Figure 2B,C). Typical images of the invasion and migration assays after *miR-30a-3p* expression are shown in Appendix A.

Although there have been many reports published on the analysis of *miR-30a-5p* (the guide strand of pre-*miR-30a*), there are few analyses of *miR-30a-3p* (the passenger strand) in the literature. Therefore, we focused on *miR-30a-3p* and explored its tumor-suppressive function and target genes in BC cells.

### 2.3. Identification of miR-30a-3p-Controlled Cancer-Promoting Genes in BC Cells

The next area of interest is determining which genes are controlled by antitumor *miR-30a-3p* in BC cells.

To identify *miR-30a-3p*-controlled genes in BC cells, we created a new gene expression profile using *miR-30a-3p*-transfected MDA-MB-231 cells by RNA sequencing. This profile revealed that 525 genes were downregulated (log 2-fold ratio < −0.7) in *miR-30a-3p*-transfected cells compared with non-transfected cells. Among these genes, 189 genes had the putative *miR-30a-3p* binding site in their 3′-untranslated region, as determined via a TargetScan Human database (release 8.0) analysis. Our strategy for *miR-30a-3p* target searching is shown in Figure 3.

Furthermore, the molecular functions of 189 genes were investigated using GeneCodis 4 databases (Appendix A). Notably, a total of 17 genes (*ANLN*, *MKI67*, *CCNB1*, *NCAPG*, *ZWINT*, *E2F7*, *PDS5A*, *RIF1*, *BIRC5*, *MAD2L1*, *CACUL1*, *KIF23*, *UBE2S*, *EML4*, *SEPT10*, *CLTC*, and *PCNP*) were found to be involved in the cell cycle (Table 1). Genes involved in cell cycle regulation are suitable as therapeutic targets for cancer cells.

### 2.4. Clinical Significance of Cell-Cycle-Related Genes Determined via TCGA-BRCA Database Analysis

Seventeen genes were subjected to clinicopathological analysis using the TCGA-BRCA dataset. Among these genes, 10 genes were significantly overexpressed in BC tissues (n = 1085) compared with normal tissues (n = 291; *p* < 0.01; Figure 4).

In addition, four genes (*ANLN*, *CCNB1*, *BIRC5*, and *KIF23*) showed statistically significant correlations with poor overall survival (10-year survival rate, *p* < 0.05; Figure 5).

### 2.5. Expression Control of Four Genes (ANLN, CCNB1, BIRC5, and KIF23) by miR-30a-3p in BC

We investigated whether four genes (*ANLN*, *CCNB1*, *BIRC5*, and *KIF23*) were controlled by *miR-30a-3p* using *miR-30a-3p*-transfected BC cells. Our data revealed that the mRNA expression levels of all genes were significantly reduced in *miR-30a-3p*-transfected BC cells, i.e., MDA-MB-157 and MDA-MB-231 (Figure 6A).

Moreover, a large number of clinical specimens were analyzed using the TCGA-BRCA database, and four genes were found to be controlled by *miR-30a-3p*. A Spearman’s rank test confirmed negative correlations between all four genes’ expression and *miR-30a-3p* expression (Figure 6B).

### 2.6. Direct Regulation of ANLN by miR-30a-3p in BC Cells

Using luciferase reporter assays, we demonstrated that *miR-30a-3p* directly binds to the 3′UTR of the *ANLN* gene. The putative *miR-30a-3p* binding site on the 3′UTR of the ANLN gene is shown in Figure 7. Luciferase activity was markedly decreased when BC cells (MDA-MB-231) were co-transfected with *miR-30a-3p* and a vector containing a *miR-30a-3p* binding sequence (Figure 7). In contrast, no decrease in luciferase activity was observed when a vector lacking the *miR-30a-3p* binding sequence was used (Figure 7). These results indicated that *miR-30a-3p* directly binds to the 3′UTR of *ANLN* and modulates its expression in BC cells.

In addition, the expression level of ANLN protein was suppressed via the ectopic expression of *miR-30a-3p* in BC tissues (Appendix A).

### 2.7. Functional Significance of ANLN in BC Cells

We investigated the oncogenic function of *ANLN* in BC cells using siRNA-mediated *ANLN* knockdown assays. The expression levels of *ANLN* were significantly reduced by two siRNAs (si*ANLN*-1 and si*ANLN*-2) in BC cells (Appendix A).

The knockdown of *ANLN* inhibited cell proliferation and markedly inhibited cell invasion and migration (Figure 8A–C). Typical images of invasion and migration assays in siRNA-transfected BC cells are shown in Appendix A.

### 2.8. Clinical Significance of ANLN in BC Clinical Specimens

Immunostaining was performed to confirm the expression of ANLN in BC clinical specimens. We observed stronger immunostaining of the ANLN protein in cancerous tissues compared to normal breast tissues (Figure 9A).

A multivariate analysis revealed that *ANLN* expression is an independent prognostic factor for BC, even when accounting for clinical prognostic factors including age, T-factor, N-factor, and M-factor (Figure 9B). Specifically, higher levels of *ANLN* expression were correlated with a poorer 10-year overall survival rate.

To identify *ANLN*-mediated molecular pathways in BC patients, we conducted a gene set enrichment analysis with TCGA-BRCA data. The “E2F targets”, “G2M checkpoint”, and “MYC target” pathways were enriched in patients with high *ANLN* expression rather than low *ANLN* expression (Table 2, Figure 9C).

## 3. Discussion

In the human genome, the *miR-30* family comprises six miRNA members (*miR-30a*, *miR-30b*, *miR-30c-1*, *miR-30c-2*, *miR-30d*, and *miR-30e*).

Two miRNAs (the guide strand and the passenger strand) are derived from each miRNA; i.e., 12 mature miRNAs are formed from the *miR-30* family [17]. Notably, all guide strands derived from the *miR-30* family have the same seed sequences. In contrast, the seed sequences of the passenger strands are divided into two groups (*miR-30a-3p*, *miR-30d-3p* and *miR-30e-3p*; and *miR-30b-3p*, *miR-30c-1-3p*, and *miR-30c-2-3p*) [17]. The tumor-suppressive function of the guide strands derived from the *miR-30* family has previously been reported in many types of cancers including BC, and various cancer-promoting genes regulated by these miRNAs have been identified [18,19,20]. Compared with the analysis of the guide strands of the *miR-30* family, the involvement of the passenger strands of the *miR-30* family in the molecular pathogenesis of BC has not been thoroughly investigated.

Based on our miRNA signature of BC, we recently investigated the *miR-30c-1-3p* and *miR-30c-2-3p* family members in BC cells [14]. A TCGA analysis showed that the low expression of these miRNAs significantly impacts the prognosis of BC patients. Functional assays demonstrated that two miRNAs acted as tumor-suppressive miRNAs in BC cells by targeting cell-cycle-related genes (e.g., *TRIP13*, *CCNB1*, *RAD51*, *PSPH*, *CENPN*, *KPNA2*, and *MXRA5*) [14]. Recently, the tumor-suppressive functions of *miR-30c-1-3p* and *miR-30c-2-3p* have been reported in multiple types of cancer, e.g., lung cancer, brain tumor, and pancreatic ductal adenocarcinoma [21,22,23].

In this study, we focused on *miR-30a-3p* and investigated its functional significance and its control of cancer-promoting genes in BC cells. Although the downregulation of *miR-30a-3p* expression was previously reported in BC tissues [24], little functional analysis of *miR-30a-3p* has been performed in BC cells. Our functional analysis demonstrated that *miR-30a-3p* is a tumor-suppressive miRNA in BC cells, similar to *miR-30c-1-3p* and *miR-30c-2-3p*. Previous reports have demonstrated that *miR-30a-3p* has antitumor functions in several types of cancer, e.g., gastric cancer, renal cell carcinoma, bladder cancer, pancreatic ductal adenocarcinoma, and lung adenocarcinoma [25,26,27,28]. We recently created an miRNA expression signature of small-cell lung cancer (SCLC), and this signature showed the downregulation of both strands of pre-*miR-30a* [29]. Our gain-of-function assays indicated that *miR-30a-3p* inhibited aggressive phenotypes of SCLC cells, e.g., cell proliferation and induced cell cycle arrest and apoptosis [29]. Previous studies and our present data have revealed that *miR-30a-3p* acts as an antitumor miRNA in various cancers, including BC. Many studies have been performed on *miR-30a-5p* (the guide strand) to date, and there is a consensus that *miR-30a-5p* has antitumor functions in BC cells [18]. In other words, it was shown that two types of miRNAs (*miR-30a-5p* and *miR-30a-3p*) derived from pre-*miR-30a* regulate various genes and are deeply involved in the molecular pathogenesis of BC.

Our next focus of interest was to identify cancer-promoting genes that are regulated by the antitumor effects of *miR-30a-3p* in BC cells. A notable feature of this analysis was that many of the genes regulated by *miR-30a-3p* in BC cells were cell-cycle-related genes. In particular, the high expression of four genes (*ANLN*, *CCNB1*, *BIRC5*, and *KIF23*) could predict the prognosis of BC patients.

Anillin (*ANLN*) functions as a scaffolding protein in coordination with actin and cytoskeletal filaments at various steps of the cell cycle [30]. *ANLN* is also known to regulate intracellular signaling cascades by controlling the activity of members of the Rho family of GTPases, e.g., RhoA, RhoG, and Rac1 [31]. Recent studies have demonstrated that *ANLN* is involved in events other than cell division, such as regulating cell–cell and cell–matrix adhesion [30,32].

Our siRNA-mediated knockdown assays demonstrated that the downregulation of *ANLN* significantly inhibited cancer cell proliferation, invasion, and migration abilities, and is strongly suggested as a cancer-promoting gene in BC cells.

A large number of studies showed that the overexpression of *ANLN* was detected in various types of cancers, e.g., pancreatic ductal adenocarcinoma, colon, lung, gastric, and hepatocellular carcinoma [33,34,35]. In BC, the inhibition of *ANLN* expression was reported to attenuate the motility and proliferation of BC cells [36,37]. Furthermore, *ANLN* knockout of BC cells induced dramatic transcriptional reprogramming, and resulted in the suppression of cancer cell stemness and trans-differentiation from mesenchymal to epithelial lineages [36]. Transcriptome reprogramming is the critical event deeply involved in cancer cell progression, metastasis, and drug resistance [38]. Notably, *ANLN* has been reported to be a gene regulated by the formation of TNBC-specific super-enhancers [39]. We believe that genes transcribed by super-enhancers play an important role in the malignant progression of BC, and that the use of *ANLN* and *ANLN*-mediated genes as therapeutic targets for BC, as well as therapeutic strategies to control these molecules, is required.

## 4. Materials and Methods

### 4.1. Cell Lines and BC Clinical Specimens

In this study, two TNBC cell lines (MDA-MB-157 and MDA-MB-231) were used. Both cell lines were obtained from Public Health England (Salisbury, UK).

This study was conducted in accordance with the guidelines of the Declaration of Helsinki and was approved by the Ethics Committee of Kagoshima University (approval number 160038 28-65; date of approval: 19 March 2021).

### 4.2. Analysis of BC Clinical Specimens Using TCGA-BRCA Database

The selection of downregulated miRNAs was based on the miRNA signature we previously created in BC clinical specimens [13].

The expression data of miRNA in BC were obtained from the following database: The Cancer Genome Atlas (TCGA) (https://www.cancer.gov/tcga, accessed on 13 March 2024), Subio Platform v1.24 (Subio Inc., Aichi, Japan).

The expression data of miRNA target genes in BC clinical specimens were obtained from the following database: GEPIA2 (http://gepia2.cancer-pku.cn/#index, accessed on 13 March 2024) [40]. The clinical significance of genes in BC was obtained from the OncoLnc database (http://www.oncolnc.org/ (accessed on 13 March 2024) [41].

### 4.3. RT-qPCR and Functional Assays of miRNAs and miRNA Target Genes in BC Cells

The procedures for RNA extraction and RT-qPCR were described in our previous studies [13,33]. The sequences of primers for SYBR Green assays are summarized in Appendix A.

Functional assays (e.g., proliferation, invasion, and migration) were performed for the transient transfection of small RNAs (miRNAs and siRNAs) into BC cell lines. The analysis procedures have been described in our previous studies [13,33]. The siRNAs and miRNAs used in the experiments are shown in Appendix A.

### 4.4. Identification of Oncogenic Targets Controlled by miR-30a-3p in BC Cells

To identify oncogenic targets controlled by *miR-30a-3p* in BC cells, we created a gene expression profile using *miR-30a-3p*-transfected MDA-MB-231 cells. We used the TargetScan Human database (https://www.targetscan.org/vert_80/ (accessed on 13 March 2024)) to identify genes that have an *miR-30a-3p* binding site in their 3′UTR.

We used the GeneCodis 4 software platform (https://genecodis.genyo.es/, accessed on 13 March 2024) to infer the molecular functions of the *miR-30a-3p* target genes [42]. Gene set enrichment analysis (GSEA) software program 4.3.2 was used to infer the molecular pathways controlled by these genes [43,44].

### 4.5. Western Blotting and Immunohistochemistry

Western blotting and immunohistochemical analysis were performed according to our previous studies [13,14]. Anti-ANLN human/mouse monoclonal IgG was used as a primary antibody. The antibodies used in this study are listed in Appendix A. A BC tissue array, BRC711 (Quickarrays Inc., CA, USA), was used for immunohistochemistry.

### 4.6. Plasmid Construction and Dual-Luciferase Reporter Assay

Vector construction and dual-luciferase reporter assays were performed as described in our previous studies [14]. The vector insertion sequences are shown in Appendix A.

### 4.7. Statistical Analysis

Statistical analyses were performed using JMP Pro 16 (SAS Institute Inc., Cary, NC, USA) and GraphPad Prism 8 (GraphPad Software, CA, USA). Differences between two groups were analyzed using Welch’s t-test, and differences between multiple groups were analyzed using Dunnett’s test. Survival rates were analyzed using Kaplan–Meier survival curves and log-rank tests.

## 5. Conclusions

Our miRNA signature and TCGA-BRCA database analysis revealed that *miR-30a-3p* (the passenger strand) was significantly downregulated in BC clinical specimens. The ectopic expression of *miR-30a-3p* attenuated the malignant phenotypes of BC cells, suggesting that this miRNA acted as an antitumor miRNA in BC cells. In total, four genes (*ANLN*, *CCNB1*, *BIRC5*, and *KIF23*) were identified as therapeutic targets via *miR-30a-3p* regulation in BC cells. *ANLN* was directly regulated by *miR-30a-3p*, and its overexpression facilitated BC cell aggressiveness. It has been suggested that *ANLN* or *ANLN*-mediated molecular pathways may be therapeutic targets for BC. Studying the involvement of the passenger strand in the molecular pathogenesis of BC and searching for its regulatory genes are effective strategies for discovering therapeutic targets for BC.

## Figures and Tables

**Figure 1 ncrna-10-00060-f001:**
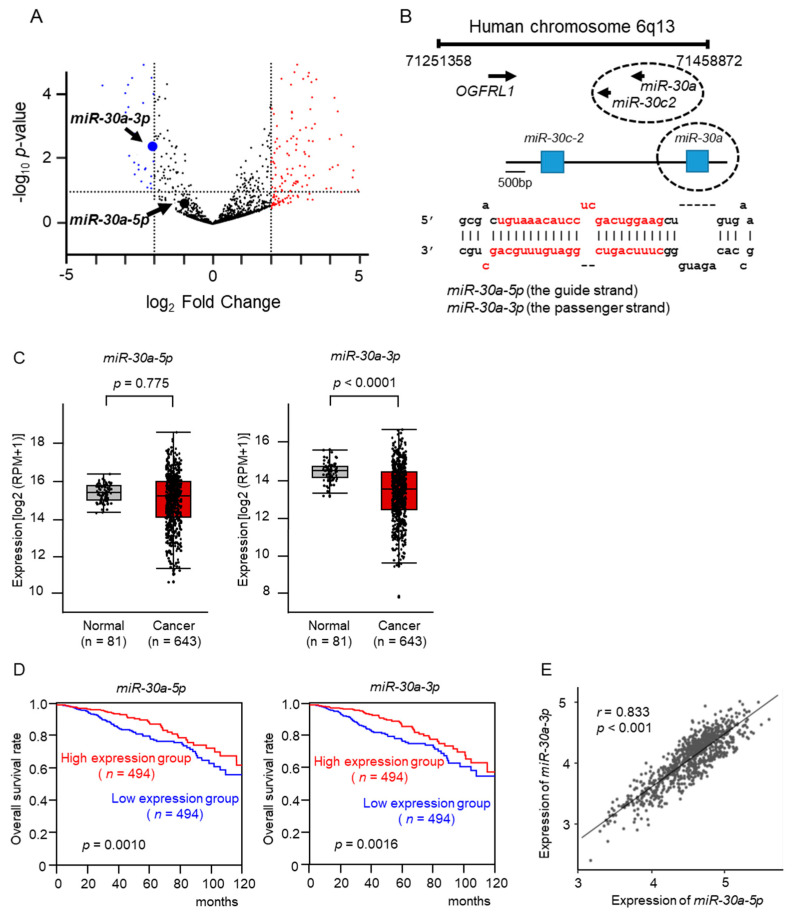
Expression levels of *miR-30a-5p* and *miR-30a-3p* in BC clinical specimens. (**A**) Volcano plot of the miRNA expression signature based on RNA sequencing (GEO accession number: GSE118539). The log_2_ fold change (FC) in the expression is plotted on the *x*-axis and the log_10_ *p*-value is on the *y*-axis. Blue and red dots represent the downregulated (log_2_FC < −2.0 and *p* < 0.05) and upregulated (log_2_FC > 2.0 and *p* < 0.05) miRNAs, respectively. (**B**) Chromosomal location of pre-*miR-30* within the human genome, showing mature sequences of *miR-30a-5p* (guide strand) and *miR-30a-3p* (passenger strand). (**C**) Expression levels of *miR-30a-5p* and *miR-30a-3p* were validated in BC clinical specimens. *miR-30a-3p* expression was significantly downregulated in cancer tissues (*p* < 0.001). (**D**) The 10-year overall survival rate of breast cancer patients according to miRNA expression. Patients with a high expression of *miR-30a-5p* and *miR-30a-3p* show a preferable prognosis. (**E**) A positive correlation (Spearman’s rank test) between *miR-30a-5p* and *miR-30a-3p* expression levels in clinical specimens is shown (*r* = 0.833, *p* < 0.001).

**Figure 2 ncrna-10-00060-f002:**
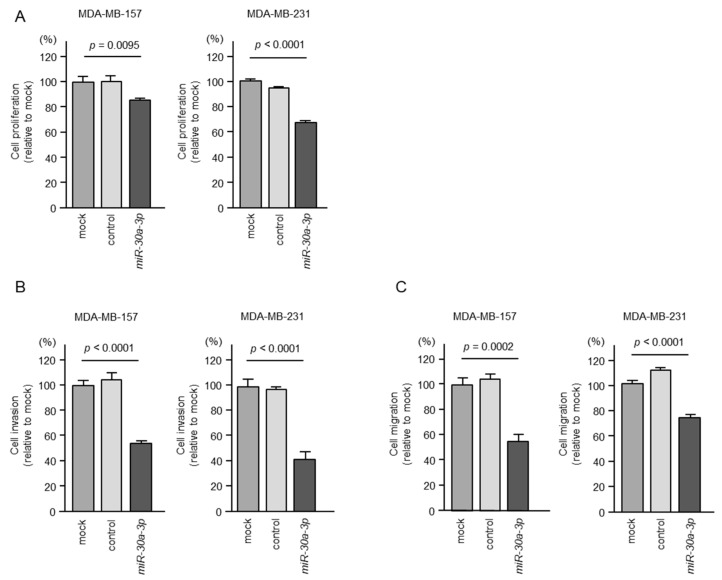
Effects of ectopic expression of *miR-30a-3p* in BC cells (MDA-MB-157 and MDA-MB-231). (**A**) Cell proliferation was assessed via the XTT assay 72 h after transient transfection of miRNAs. (**B**) Cell invasion was evaluated using Matrigel invasion assays 48 h after *miR-30a-3p*-transfected cells were seeded into the chambers. (**C**) Cell migration was evaluated using a membrane culture system 48 h after *miR-30a-3p*-transfected cells were seeded into chambers.

**Figure 3 ncrna-10-00060-f003:**
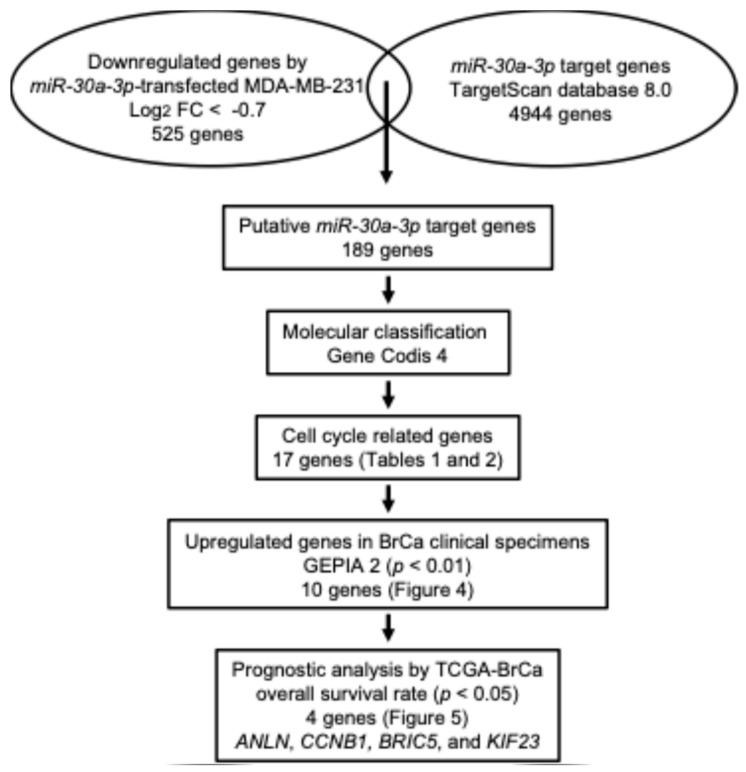
Flowchart of *miR-30a-3p* target identification in BC cells. To identify putative targets of *miR-30a-3p* in BC cells, we integrated two datasets: the TargetScan Human database (release 8.0) and our original mRNA expression profile (*miR-30a-3p*-transfected MDA-MB-231 cells; GEO accession number: GSE118539). A total of 189 genes were identified as putative *miR-30a-3p* targets. According to a GeneCodis 4 database analysis, 17 genes were most frequently associated with the cell cycle. Of these 17 genes, 10 genes showed significant overexpression in BC specimens according to GEPIA2 (http://gepia2.cancer-pku.cn/#index, accessed on 13 March 2024), and 4 genes showed a significantly low overall survival rate according to OncoLnc (http://www.oncolnc.org/, accessed on 13 March 2024). Finally, four oncogenic genes were selected as *miR-30a-3p* targets in BC.

**Figure 4 ncrna-10-00060-f004:**
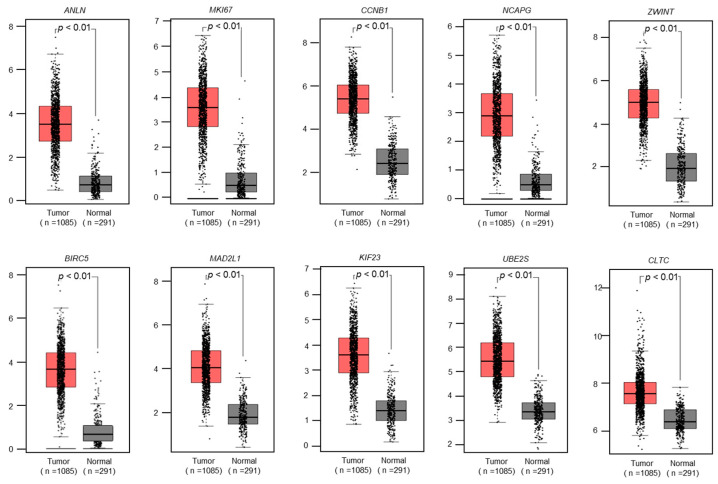
The expression levels of 10 genes (*ANLN*, *MKI67*, *CCNB1*, *NCAPG*, *ZWINT*, *BIRC5*, *MAD2L1*, *KIF23*, *UBE2S*, and *CLTC*) in BC clinical specimens were analyzed using TCGA-BRCA datasets. These genes were upregulated in BC tissues (n = 1085) compared with normal tissues (n = 291) (*p* < 0.01).

**Figure 5 ncrna-10-00060-f005:**
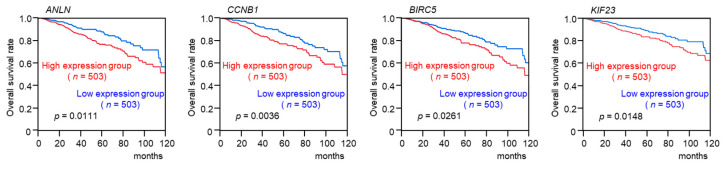
Clinical significance of 4 target genes in BC specimens. Kaplan–Meier curves of 10-year overall survival rates according to the levels of gene expression (*ANLN*, *CCNB1*, *BIRC5*, and *KIF23*). A total of 1006 patients were divided into high- and low-expression groups according to the median gene expression level. The red lines represent the high-expression group, and the blue lines represent the low-expression group. Expressing high levels of these genes was significantly correlated with a poorer prognosis in BC patients.

**Figure 6 ncrna-10-00060-f006:**
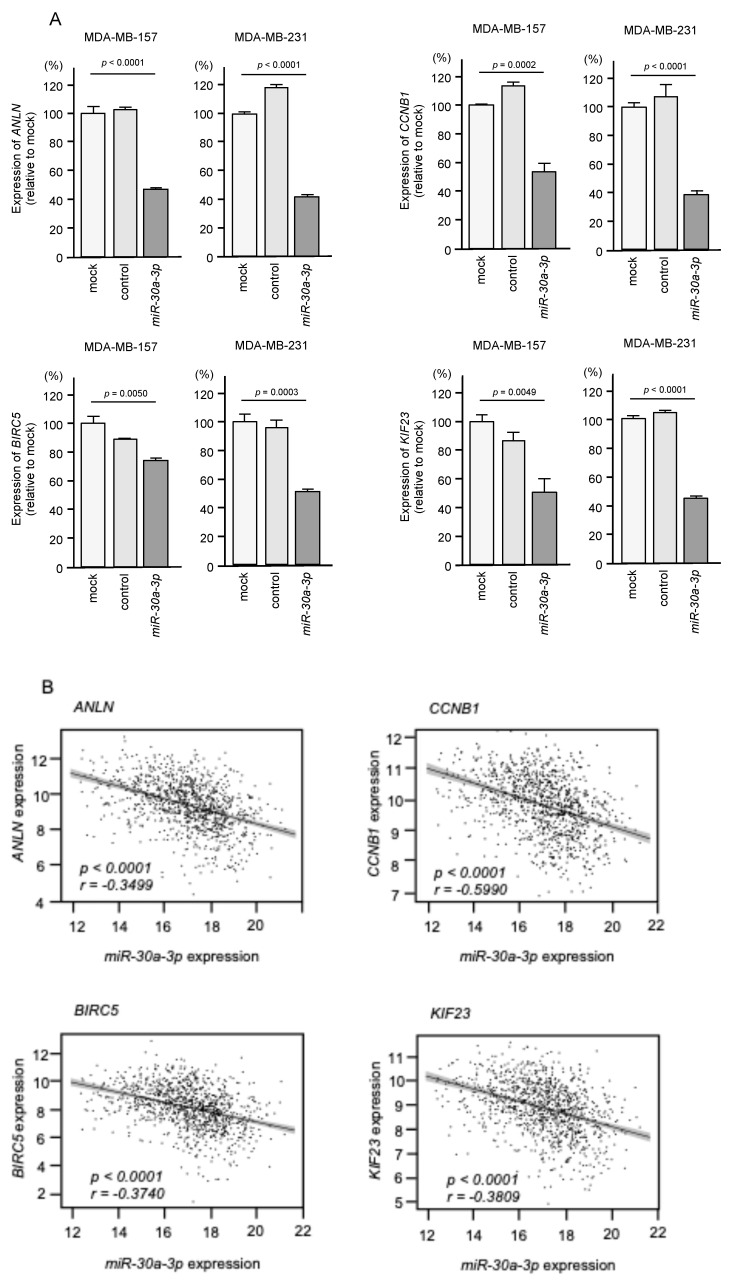
Expression control of four genes (*ANLN*, *CCNB1*, *BIRC5*, and *KIF23*) by *miR-30a-3p* in BC cells. (**A**) qRT-PCR showing significantly reduced expression of all four mRNAs 72 h after *miR-30a-3p* transfection in MDA-MB-157 and MDA-MB-231 cells compared to the control group. (**B**) Correlation analysis of four genes using TCGA-BRCA database. Expression levels of *miR-30a-3p* and four genes (*ANLN*, *CCNB1*, *BIRC5*, and *KIF23*) in BC clinical samples are negatively correlated.

**Figure 7 ncrna-10-00060-f007:**
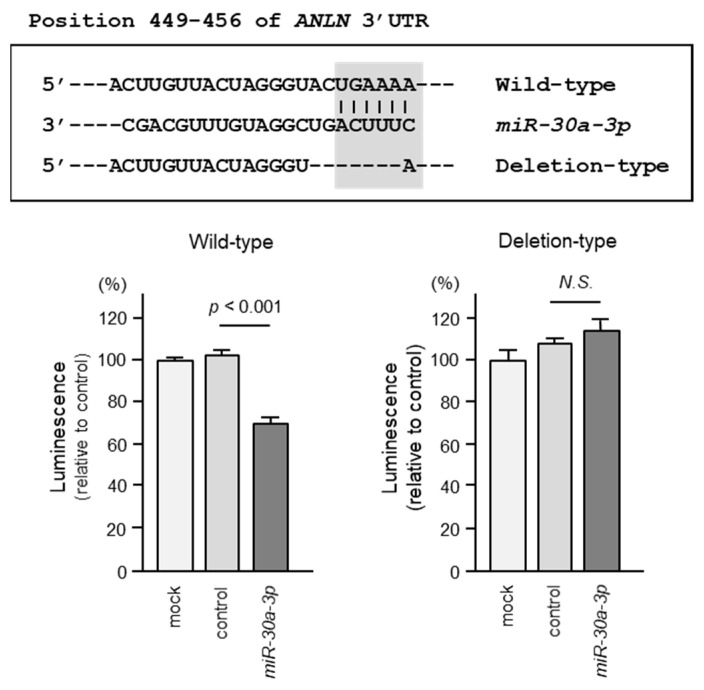
TargetScan Human database (release 8.0) shows that the putative *miR-30a-3p* binding site is mapped in the 3′UTR of the *ANLN* gene. Dual-luciferase reporter assays revealed reduced luminescence activity after co-transfection of *miR-30a-3p* with a vector containing the *miR-30a-3p* binding site (wild-type) in MDA-MB-231 cells. In contrast, no luminescence activity was observed after co-transfection of *miR-30a-3p* with a vector lacking the *miR-30a-3p* binding site (deletion-type) in MDA-MB-231 cells.

**Figure 8 ncrna-10-00060-f008:**
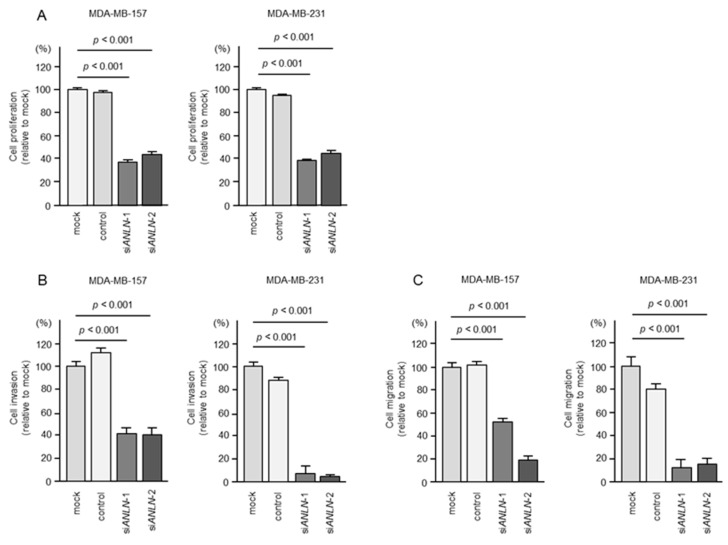
Effects of knockdown of *ANLN* by siRNAs in BC cells. (**A**) Cell proliferation was assessed using XTT assays 72 h after siRNA transfection into BC cells. (**B**) Cell invasion was evaluated using Matrigel invasion assays 48 h after si*ANLN*-transfected cells were seeded into chambers. (**C**) Cell migration was evaluated using a membrane culture system 48 h after si*ANLN*-transfected cells were seeded into chambers.

**Figure 9 ncrna-10-00060-f009:**
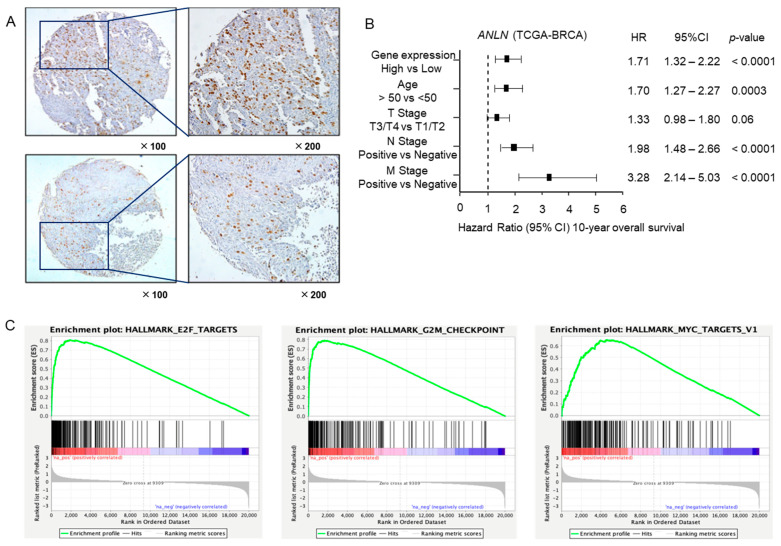
Clinical significance of *ANLN* expression in BC. (**A**) Expression of ANLN in BC tissues. Immunohistochemical staining of ANLN was confined to cancer tissues, whereas weak staining was observed in the noncancerous area. Upper: 50-year-old woman, T1N0M0 invasive papillary carcinoma. Lower: 78-year-old woman, T4N0M0 invasive breast ductal carcinoma. (**B**) Forest plot showing the result of multivariate Cox proportional hazards regression analysis of the 10-year overall survival rate. Patients with high *ANLN* expression had a significantly lower overall survival rate. These data were obtained from TCGA-BRCA datasets. (**C**) Gene set enrichment analysis (GSEA) was applied to explore molecular pathways mediated by *ANLN* in BC cells. The top three pathways enriched in BC patients with high *ANLN* expression were E2F targets, G_2_M checkpoint, and Myc targets.

**Table 1 ncrna-10-00060-t001:** Cell-cycle-related genes affected by *miR-30a-3p* regulation in BC cells.

Entrez Gene ID	Gene Symbol	Gene Name	Total Binding Sites	Log2 Fold Change
54443	*ANLN*	anillin, actin binding protein	1	−1.7064619
4288	*MKI67*	antigen identified by monoclonal antibody Ki-67	1	−1.2917953
891	*CCNB1*	cyclin B1	1	−1.2548676
64151	*NCAPG*	non-SMC condensin I complex, subunit G	1	−1.0997882
11130	*ZWINT*	ZW10 interacting kinetochore protein	1	−1.0036101
144455	*E2F7*	E2F transcription factor 7	1	−0.9913907
23244	*PDS5A*	PDS5, regulator of cohesion maintenance, homolog A (*S. cerevisiae*)	3	−0.9755006
55183	*RIF1*	RAP1 interacting factor homolog (yeast)	3	−0.9548717
332	*BIRC5*	baculoviral IAP repeat containing 5	1	−0.9173002
4085	*MAD2L1*	MAD2 mitotic arrest deficient-like 1 (yeast)	2	−0.9139371
143384	*CACUL1*	CDK2-associated, cullin domain 1	4	−0.8496602
9493	*KIF23*	kinesin family member 23	1	−0.8479867
27338	*UBE2S*	ubiquitin-conjugating enzyme E2S	1	−0.8329787
27436	*EML4*	echinoderm microtubule associated protein like 4	1	−0.8073831
151011	*SEPT10*	septin 10	2	−0.7851834
1213	*CLTC*	clathrin, heavy chain (Hc)	1	−0.755415
57092	*PCNP*	PEST proteolytic signal-containing nuclear protein	3	−0.7289238

**Table 2 ncrna-10-00060-t002:** *ANLN*-mediated pathways via gene set enrichment analysis (GSEA).

Pathway	Enrichment Score	Normalized Enrichment Score	*p*-Value	FDR
HALLMARK_E2F_TARGETS	0.81	3.39	<0.001	<0.001
HALLMARK_G2M_CHECKPOINT	0.79	3.31	<0.001	<0.001
HALLMARK_MYC_TARGETS_V1	0.65	2.72	<0.001	<0.001
HALLMARK_MITOTIC_SPINDLE	0.6	2.5	<0.001	<0.001
HALLMARK_MTORC1/SIGNALING	0.59	2.48	<0.001	<0.001

## Data Availability

The data presented in this study are available on request from the corresponding author.

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
