# Peer review of "Identification of Tumor-Suppressive *miR-30a-3p* Controlled Genes: *ANLN* as a Therapeutic Target in Breast Cancer"

_ncrna, 2024, doi:10.3390/ncrna10060060_

Round 1
Reviewer 1 Report
Comments and Suggestions for Authors
Mitsueda et al. did a deep investigation on the relationship of miR-30a and breast cancer. The authors provided both clinical data and in vitro data on miR-30a and revealed the down-stream targets of miR-30a. However, there are quite a few issues which I concern.
1. I see in Figure 1D and Figure 5, the Kaplan-Meier analysis of miR-30a and ANLN, CCNB1, BIRC5, and KIF23. The authors should detail the clinical design, enrolled patients and 10-year follow-up data. The correlation between miRNA and breast cancer prognosis is a significant point in the manuscript.
2. For the in vitro analysis, only bar graph is not enough. The authors should provide representative cell images especially for invasion and migration.
3. The authors applied miRNA mimic transfection without miRNA inhibitor transfection. Have the authors ever tried miRNA inhibitor on in vitro evaluation.
4. Only qPCR and Luciferase assay is not enough. The authors should add western blot analysis when investigating the down-stream target of miR-30a
Comments on the Quality of English LanguageThe English could be improved to more clearly express the research.
Author Response
Revise letter
Manuscript ID: ncrna-3237791
November 4, 2024
Dr. Luca Falzone
Guest Editor
Dear Dr. Falzone:
We would like to express our gratitude for your consideration of our above-mentioned manuscript for publication in non-coding RNA. Enclosed, please find the revised manuscript (ncrna-3237791) along with a detailed explanation of the revisions, which were made based on the reviewers’ comments. All changes are highlighted in the revised manuscript.
Reviewer #1
Mitsueda et al. did a deep investigation on the relationship of miR-30a and breast cancer. The authors provided both clinical data and in vitro data on miR-30a and revealed the down-stream targets of miR-30a. However, there are quite a few issues which I concern.
Comment-1: I see in Figure 1D and Figure 5, the Kaplan-Meier analysis of miR-30a and ANLN, CCNB1, BIRC5, and KIF23. The authors should detail the clinical design, enrolled patients and 10-year follow-up data. The correlation between miRNA and breast cancer prognosis is a significant point in the manuscript.
Response: All breast cancer samples information is stored in a public database. We do not have access to any information other than that provided by the database. The Kaplan-Meier analysis was not performed from clinical research data. As mentioned in Materials and Methods 4.2., the survival data was extracted from Oncolnc, an online tool that links The Cancer Genome Atlas survival data to mRNA and miRNA expression levels.
4.2. Analysis of BC Clinical Specimens by TCGA-BRCA Database
The selection of downregulated miRNAs was based on the miRNA signature we previously created in BC clinical specimens [13]. The expression data of miRNA in BC was obtained from the following databases: The 337 Cancer Genome Atlas (TCGA) (https://www.cancer.gov/tcga, accessed on 13 March 2024), Subio Platform v1.24 (Subio, Inc., Japan). Expression data of miRNA target genes in BC clinical specimens was obtained from following database: GEPIA2 (http://gepia2.cancer-pku.cn/#index (accessed on 13 March 341 2024) [40]. The clinical significance of genes in BC was obtained from OncoLnc database (http://www.oncolnc.org/ (accessed on 13 March 2024)[41].
Comment-2: For the in vitro analysis, only bar graph is not enough. The authors should provide representative cell images especially for invasion and migration.
Response: Following the reviewer's instructions, typical photographs of invasion and migration assays are presented in Supplemental Figures 2 and 4.
Comment-3: The authors applied miRNA mimic transfection without miRNA inhibitor transfection. Have the authors ever tried miRNA inhibitor on in vitro evaluation.
Response: Expression levels of miR-30a-3p was extremely low in BrCa cell lines, MDA-MB-157 and MDA-MB231. Therefore, experiments to suppress miR-30a-3p expression in BrCa cell lines using miR-30a-3p inhibitor is not feasible. Thank you for understanding.
Comment-4: Only qPCR and Luciferase assay is not enough. The authors should add western blot analysis when investigating the down-stream target of miR-30a
Response: I agree with reviewer’s suggestion, and WB analysis was performed. We confirmed that ANLN was suppressed by miR-30a-3p in BrCa cells, MDA-MB-231. Supplemental Figure has been revised and WB data added.
Thank you for your constructive comments and suggestions. Some experiments could not be performed on the reviewer's comments. However, I have made some corrections as pointed out by the reviewers, and the quality is comparable to the papers that have been accepted in your journal “non-coding RNA” so far. Again, thank you for your consideration of our manuscript for publication in your journal.
Sincerely,
Naohiko Seki, PhD
Department of Functional Genomics
Graduate School of Medicine, Chiba University
1-8-1 Inohana, Chuo-ku
Chiba 260-8670, Japan
Email: naoseki@faculty.chiba-u.jp
Reviewer 2 Report
Comments and Suggestions for Authors
The study identifies that miR-30a-3p is significantly downregulated in breast cancer tissues, with 189 potential target genes identified, 17 of which are involved in cell cycle regulation. This suggests that miR-30a-3p plays a crucial role in controlling cell proliferation in breast cancer cells. While the correlation between miR-30a-3p expression and patient prognosis is not fully elucidated, its downregulation may indicate poor outcomes. Notably, ANLN, one of the target genes of miR-30a-3p, is implicated in breast cancer cell proliferation, underscoring its significance in tumor growth and potential therapeutic approaches. The study presents interesting findings regarding the downregulation of miR-30a-3p in breast cancer tissues and its potential tumor-suppressive functions. However, I have several concerns and suggestions that should be addressed to strengthen the manuscript:
1. The study’s sample size appears limited, and expanding the cohort to include a more diverse population in terms of age, ethnicity, and breast cancer subtypes is recommended. This would not only improve the generalizability of the findings but also enhance the significance of miR-30a-5p and miR-30a-3p expression in breast cancer. Additionally, clarifying why miR-30a-5p and miR-30a-3p are expected to be correlated in Fig. 1E is important. It is also worth exploring why only miR-30a-3p was chosen for this study despite the potential relevance of both miRNAs.
2. Although the study highlights the tumor-suppressive role of miR-30a-3p through in vitro assays, the lack of in vivo validation limits the strength of the findings. To reinforce the conclusions, I recommend conducting experiments in animal models to validate the functional relevance of miR-30a-3p in breast cancer progression.
3. The manuscript would benefit from a more in-depth exploration of the mechanisms through which miR-30a-3p exerts its effects. Identifying specific signaling pathways and molecular interactions involved in miR-30a-3p regulation could provide valuable insights into its role in breast cancer biology and its potential as a therapeutic target.
4. It is crucial to account for potential confounding factors that may influence miRNA expression and patient outcomes, such as treatment history, tumor microenvironment, and other genetic or epigenetic alterations. Addressing these factors would strengthen the reliability of the study’s conclusions and ensure a more accurate interpretation of the data.
5. While the focus on miR-30a-3p is important, breast cancer is regulated by a complex network of miRNAs and other regulatory molecules. Expanding the analysis to include other members of the miR-30 family or additional relevant miRNAs could provide a more comprehensive view of the regulatory landscape and the broader implications of miR-30a-3p in breast cancer.
6. The statistical methods used to correlate miR-30a-3p expression with clinical outcomes should be clearly described and thoroughly justified. Ensuring adequate statistical power and employing appropriate statistical techniques are crucial for the validity and robustness of the findings. Additionally, providing clear legends for the IHC images would enhance clarity and interpretation.
7. While the study identifies a correlation between low miR-30a-3p expression and poor prognosis, the clinical relevance of targeting miR-30a-3p for therapeutic purposes requires further exploration. A more detailed discussion of the potential therapeutic implications of these findings would be valuable for understanding how this miRNA might be used in future treatment strategies for breast cancer.
8. Finally, the manuscript should acknowledge the possibility of publication bias within the miR-30a-3p literature on breast cancer. Including a balanced discussion that considers both positive and negative findings from other studies would offer a more comprehensive view of the current state of research and help contextualize the study's contributions.
Author Response
Revise letter
Manuscript ID: ncrna-3237791
November 3, 2024
Dr. Luca Falzone
Guest Editor
Dear Dr. Falzone:
We would like to express our gratitude for your consideration of our above-mentioned manuscript for publication in non-coding RNA. Enclosed, please find the revised manuscript (ncrna-3237791) along with a detailed explanation of the revisions, which were made based on the reviewers’ comments. All changes are highlighted in the revised manuscript.
Reviewer #2
The study identifies that miR-30a-3p is significantly downregulated in breast cancer tissues, with 189 potential target genes identified, 17 of which are involved in cell cycle regulation. This suggests that miR-30a-3p plays a crucial role in controlling cell proliferation in breast cancer cells. While the correlation between miR-30a-3p expression and patient prognosis is not fully elucidated, its downregulation may indicate poor outcomes. Notably, ANLN, one of the target genes of miR-30a-3p, is implicated in breast cancer cell proliferation, underscoring its significance in tumor growth and potential therapeutic approaches. The study presents interesting findings regarding the downregulation of miR-30a-3p in breast cancer tissues and its potential tumor-suppressive functions. However, I have several concerns and suggestions that should be addressed to strengthen the manuscript:
Comment-1: The study’s sample size appears limited, and expanding the cohort to include a more diverse population in terms of age, ethnicity, and breast cancer subtypes is recommended. This would not only improve the generalizability of the findings but also enhance the significance of miR-30a-5p and miR-30a-3p expression in breast cancer. Additionally, clarifying why miR-30a-5p and miR-30a-3p are expected to be correlated in Fig. 1E is important. It is also worth exploring why only miR-30a-3p was chosen for this study despite the potential relevance of both miRNAs.
Response: Following the reviewer's instructions, miR-30a-5p and miR-30a-3p expressions were analyzed by subtypes, luminal, HER2-positive, and TNBC. Data are presented in the Supplemental Figure 1.
The following sentence has been added in Results 2.1.
Furthermore, we investigated the expression levels of miR-30a-5p and miR-30a-3p according to BC subtypes, Luminal, HER2-positive, and TNBC. The expression level of miR-30a-5p was significantly decreased in HER2-positive and TNBC compared with normal breast tissue (Figure S1). On the other hand, decreased expression of miR-30a-3p was confirmed in all subtypes (Figure S1).
The following statement has been added as the reason for selecting miR-30a-3p in Results 2.2.
Although there have been many reports on the analysis of miR-30a-5p (the guide strand of pre-miR-30a), there have been few analyses of miR-30a-3p (the passenger strand). Therefore, we focused on miR-30a-3p, and explored its tumor suppressive function and target genes in BC cells.
Comment-2: Although the study highlights the tumor-suppressive role of miR-30a-3p through in vitro assays, the lack of in vivo validation limits the strength of the findings. To reinforce the conclusions, I recommend conducting experiments in animal models to validate the functional relevance of miR-30a-3p in breast cancer progression.
Response: We recognize that reviewer comments (in vivo assays) are important in cancer research. However, in this study, we aimed to find the target molecules of microRNAs through in vitro analysis. Unfortunately, there were no small molecule compounds that could inhibit the target molecule, ANLN. In this study, we do not analyze in vivo assays. We appreciate your understanding
Comment-3: The manuscript would benefit from a more in-depth exploration of the mechanisms through which miR-30a-3p exerts its effects. Identifying specific signaling pathways and molecular interactions involved in miR-30a-3p regulation could provide valuable insights into its role in breast cancer biology and its potential as a therapeutic target.
Response: In this study, we identified antitumor miR-30a-3p controlled genes by genome-wide gene expression analysis. Interestingly, many of the genes controlled by miR-30a-3p are cell cycle-related genes (Table 1). Furthermore, it has been suggested that the molecular pathways regulated by ANLN are involved in the cell cycle and cell division (Figure 9 and Table 2). Based on these findings, we believe that it is well worth exploring ANLN as a therapeutic target for breast cancer.
We added the following sentence in 5. Conclusion.
It has been suggested that ANLN or ANIN-mediated molecular pathways may be therapeutic targets for BC.
Comment-4: It is crucial to account for potential confounding factors that may influence miRNA expression and patient outcomes, such as treatment history, tumor microenvironment, and other genetic or epigenetic alterations. Addressing these factors would strengthen the reliability of the study’s conclusions and ensure a more accurate interpretation of the data.
Response: Since these data are extracted from The Cancer Genome Atlas, information on treatment history, tumor microenvironment, and genetic or epigenetic alterations are hardly collected.
Comment-5: While the focus on miR-30a-3p is important, breast cancer is regulated by a complex network of miRNAs and other regulatory molecules. Expanding the analysis to include other members of the miR-30 family or additional relevant miRNAs could provide a more comprehensive view of the regulatory landscape and the broader implications of miR-30a-3p in breast cancer.
Response: The following sentence has been added in Introduction (last chapter).
Based on our miRNA signature of BC, we focused on miR-30-family whose expressions were suppressed in BC tissues. We have continued to explore the antitumor functions of miR-30-family members (miR-30c-1-3p and miR-30c-2-3p), and their target molecules in BC [13,14].
Comment-6: The statistical methods used to correlate miR-30a-3p expression with clinical outcomes should be clearly described and thoroughly justified. Ensuring adequate statistical power and employing appropriate statistical techniques are crucial for the validity and robustness of the findings. Additionally, providing clear legends for the IHC images would enhance clarity and interpretation.
Response: Thank you for pointing out the statistical analysis in this study. We reviewed the statistical analysis again but found no particular problems with the analysis. We added the clinical information of IHC images in the Figure legend of Figure 9 as follows: Upper: 50-year-old woman, T1N0M0 invasive papillary carcinoma. Lower: 78-year-old woman, T4N0M0 invasive breast ductal carcinoma.
Comment-7: While the study identifies a correlation between low miR-30a-3p expression and poor prognosis, the clinical relevance of targeting miR-30a-3p for therapeutic purposes requires further exploration. A more detailed discussion of the potential therapeutic implications of these findings would be valuable for understanding how this miRNA might be used in future treatment strategies for breast cancer.
Response: The following sentence has been added in last chapter of Discussion.
Interestingly, ANLN has been reported to be a gene regulated by the formation of TNBC-specific super-enhancer [39]. We believe that genes transcribed by super-enhancer play an important role in the malignant progression of BC. ANLN and ANLN-mediated genes as therapeutic targets for BC, and therapeutic strategies to control these molecules is required.
Comment-8: Finally, the manuscript should acknowledge the possibility of publication bias within the miR-30a-3p literature on breast cancer. Including a balanced discussion that considers both positive and negative findings from other studies would offer a more comprehensive view of the current state of research and help contextualize the study's contributions.
Response: Thank you for pointing that out. We searched for previous studies on miR-30a-3p in cancer cells, but found no papers demonstrating the cancer-promoting functions of miR-30a-3p.
Thank you for your constructive comments and suggestions. Some experiments could not be performed on the reviewer's comments. However, I have made some corrections as pointed out by the reviewers, and the quality is comparable to the papers that have been accepted in your journal “non-coding RNA” so far. Again, thank you for your consideration of our manuscript for publication in your journal.
Sincerely,
Naohiko Seki, PhD
Department of Functional Genomics
Graduate School of Medicine, Chiba University
1-8-1 Inohana, Chuo-ku
Chiba 260-8670, Japan
Email: naoseki@faculty.chiba-u.jp
Round 2
Reviewer 2 Report
Comments and Suggestions for Authors
This manuscript can be approved for publication